# Perceived causes of marital dissatisfaction among Nigerian immigrants in North America: A qualitative study

Jochebed Ade-Oshifogun 1*⊙, Augusta Olaore2⊙, Kwabena Adu Agyemang3⊙

1 School of Nursing, College of Health, Education and Social Transformation, New Mexico State University, Las Cruces, New Mexico, United States of America, 2 Department of Social Work and Community Development, University of Johannesburg, Johannesburg, South Africa, 3 Department of Computer Science, College of Arts and Social Sciences, New Mexico State University, Las Cruces, New Mexico, United States of America

⊙ These authors contributed equally to this work.

* Jade7@nmsu.edu

## Abstract

Marital dissatisfaction among Nigerian immigrants in North America (NINA) arises from a complex interplay of cultural transitions, acculturation stress, and socio-economic pressures. This cross-sectional study followed a phenomenological approach and the guidelines from the Consolidated criteria for reporting qualitative research (COREQ) to examine the perceived causes of marital dissatisfaction among NINA through in-depth interviews with 15 participants residing in the United States and Canada. Participants were adults with at least one year of living with a Nigerian spouse in North America. Ten themes emerged from the data analysis identifying key marital challenges faced by Nigerian immigrant couples in North America. These include cultural conflicts between patriarchal and egalitarian values, financial stress, job insecurity, and extended family obligations. Other challenges involve social norms discouraging open discussions, experiences of abuse, lack of relationship skills, peer and societal pressures, infidelity, parenting conflicts, immigration-related stress, and an overreliance on prayer without practical interventions. These findings highlight the need for culturally sensitive support systems for this population and policies focusing on providing culturally tailored marital counseling, financial literacy programs, and accessible culturally sensitive mental health support services.

## Introduction

Marriage is a fundamental institution shaping societal structures worldwide. Marital satisfaction is a dynamic construct influenced by individual, interpersonal, and societal factors, including personality traits, spirituality, communication, intimacy, and mental health. Research highlights the impact of socio-demographic variables such as occupation, marriage duration, age, number of children, and financial

**Data availability statement:** The data underlying the findings of this study are included within the manuscript. Due to ethical and confidentiality considerations, complete interview transcripts are not publicly available. However, de-identified excerpts or additional data may be made available upon reasonable request to the corresponding author, pending approval from the institutional ethics committee. Researchers who meet the criteria for access to confidential data can contact the Andrews University Institutional Data Access via irb@andrews.edu.

**Funding:** Grant received by JBA Grant number:11-2011-xxxx-75-201172 Funder: Andrews University, Berrien Springs, MI Faculty Research Grant https://www.andrews.edu/services/research/faculty_resources/faculty_research_grants/index.html This study was supported by the Faculty Research Grant from Andrews University, Berrien Springs, Michigan, USA. The funder had no role in the study design, data collection and analysis, decision to publish, or preparation of the manuscript.

**Competing interests:** The authors have declared that no competing interests exist.

stability on marital satisfaction. The experience of Nigerian immigrants in North America (NINA) offers a unique lens through which to examine the challenges that arise in marital relationships within the context of cultural adaptation, acculturation, and socio-economic shifts. Unlike marital satisfaction, which has been widely studied, marital dissatisfaction—the precursory stage to marital instability and potential dissolution—remains underexplored, particularly within the NINA population. Understanding the perceived causes of marital dissatisfaction is essential for the prevention of marital disruption and the promotion of healthier relationships within this immigrant community.

The transition from Nigerian cultural norms to North American (NA) societal expectations presents distinct challenges for Nigerian immigrants, particularly regarding gender roles, financial expectations, and support systems. Acculturation stress, shifting gender dynamics, financial instability, and social isolation have been identified as key contributors to marital dissatisfaction among immigrant populations [1,2]. Nigerian culture, predominantly patriarchal, enforces traditional gender roles, which often clash with the more egalitarian norms prevalent in NA. The revised National Gender Policy of Nigeria (2021) highlights how patriarchal structures perpetuate gender-based inequalities, reinforcing male dominance and female subjugation. While these norms remain entrenched within Nigerian communities, the NA environment challenges these conventions, often leading to conflicts within marital relationships [3].

Furthermore, Nigerian culture is deeply rooted in communal support structures that facilitate financial and emotional well-being. The collectivist nature of Nigerian society ensures that familial and community networks provide economic and social support, including assistance with childcare, financial obligations, and conflict mediation [4]. The absence of these traditional support networks in NA exacerbates marital strain, particularly for Nigerian immigrant women who may struggle to balance cultural expectations with newfound societal norms [5]. Additionally, economic pressures—including the expectation of financial remittances to extended family in Nigeria—place further stress on marital relationships. Reports from the World Bank indicate that Nigerian immigrants significantly contribute to remittances, amounting to $19.55 billion in 2022, underscoring the financial burdens that can strain marital harmony [6].

The economic challenges faced by Nigerian immigrants in NA are compounded by employment disparities, underemployment, and racial bias in hiring practices [7]. Many Nigerian immigrants experience deskilling due to the non-recognition of foreign qualifications, limiting their employment opportunities and exacerbating financial strain. Studies have linked financial hardship with marital dissatisfaction, citing economic instability as a leading cause of stress and discord among immigrant couples [2]. Additionally, power imbalances in marital relationships often manifest in financial dependency, reinforcing patriarchal dynamics and contributing to intimate partner violence (IPV) [7].

A growing body of literature examines marital dissatisfaction, yet most studies focus on marital dissolution, particularly divorce, rather than dissatisfaction as a persistent issue [1]. Cultural and economic factors often deter Nigerian immigrants from

pursuing divorce, leading many to remain in unhappy marriages due to familial obligations, financial concerns, or religious beliefs [2]. However, since marital dissatisfaction frequently precedes divorce, it remains a critical area of study [7].

Marital dissatisfaction in this study refers to any ongoing negative perception or experience within the marital relationship that undermines emotional and physical well-being, fulfillment, or a sense of connection. It may involve emotional disconnection, unmet expectations, gender-role conflicts, poor communication, or cumulative stress resulting from interpersonal dynamics or structural factors [8,9].

## Existing research and identified gap

Existing research highlights various factors influencing marital instability among Nigerian immigrants, yet a crucial gap remains in understanding the early development of dissatisfaction. Gbadegesin and Adefi link patriarchal norms in Nigerian immigrant marriages to domestic violence and marital discord but do not explore how dissatisfaction begins and evolves over time [10]. Similarly, Ekwemalor and Ezeobele examine immigration-related stress and its effects on mental health and family dynamics but overlook how these stressors contribute to persistent marital dissatisfaction [11]. Sheykhi analyzes global divorce trends shaped by modernization but does not address the underlying dissatisfaction that precedes separation [12]. Additionally, Sbarra et al. focus on the psychological and emotional consequences of divorce, yet their emphasis remains on post-divorce outcomes rather than the progression of dissatisfaction that leads to marital breakdown [13].

Existing research primarily examines marital discord, divorce, and post-divorce consequences while neglecting the gradual buildup of dissatisfaction that precedes separation. This study shifts the focus from dissolution to dissatisfaction, exploring how Nigerian immigrants navigate marital discord in a foreign cultural landscape. Given these gaps in the literature, there is a pressing need for research that examines the early signs and progression of marital dissatisfaction among Nigerian immigrant couples. Understanding the intersection of cultural expectations, immigration stress, and relationship dynamics can provide valuable insights into preventive strategies and interventions. This study aims to fill that gap by exploring the nuanced experiences of marital dissatisfaction within this population before it escalates into divorce or separation.

## Rationale for study

Most research on marital satisfaction and dissatisfaction centers on Western populations, with limited studies on African immigrants in North America. Studies on Nigerian marital relationships largely explore gender roles, patriarchy, and financial expectations within Nigeria, but fail to examine how these dynamics shift after migration [4,5]. While acculturation studies address immigrant family structures, there is inadequate understanding of its specific impact on marital dissatisfaction among Nigerian immigrants.

Furthermore, research on marital instability often emphasizes divorce and legal separations while overlooking sustained dissatisfaction without dissolution [1]. Many Nigerian immigrant couples, bound by cultural norms and economic constraints, remain in dissatisfactory marriages despite experiencing significant marital strain [2]. The persistence of marital distress without separation highlights the need to study dissatisfaction as an independent issue.

This study fills the gap by exploring perceived causes of marital dissatisfaction among Nigerian immigrants in NA. The decision to focus on perceived rather than lived experiences was based on the understanding that, within immigrant communities, experiences are often shared collectively, and social observations—especially within extended families and community networks—significantly shape individual perceptions of marital dynamics. As Uzun noted, "Central to these diverse immigrant communities is the role of collective memory in preserving their distinct identities, histories, and shared experiences.[14]" This approach also reduces the risk of stigmatization, as participants may feel more comfortable discussing general observations than personal experiences. Cho highlights that migration trauma is inherently collective and intercultural, reinforcing the need for culturally sensitive methodologies [15]. Thus, by centering on perceptions, this study

aligns with cultural norms that may discourage direct disclosure of marital dissatisfaction, while still yielding meaningful insights [16].

Through a qualitative approach, it captures perceived lived experiences, shedding light on the cultural, economic, and interpersonal factors contributing to marital distress. Findings will inform culturally sensitive interventions to mitigate dissatisfaction and promote healthier relationships within this population.

## Methods

### Study design

This study utilized a cross-sectional qualitative research approach to investigate the perceived causes of marital dissatisfaction among Nigerian immigrants in NA. This qualitative study used a semi-structured interview guide to explore participants' perceptions and experiences of the factors contributing to marital dissatisfaction among Nigerian immigrants in NA. This phenomenological approach explores the observed lived experiences and their meanings through an individual's perspective [17]. This significantly captures the essence of people's experiences through a reflective and interpretive method. Van Manen's methodology combines existential and hermeneutic dimensions, facilitating a deep understanding of personal narratives and their meanings [17]. This phenomenological approach explores the observed lived experiences and their meanings through an individual's perspective [17]. This significantly captures the essence of people's experiences through a reflective and interpretive method. .

### Participants

A purposive sampling approach was used to recruit participants from the United States and Canada based on the following inclusion criteria: (a) Nigerian immigrants residing in NA; (b) currently or previously married to another Nigerian immigrant in NA; (c) proficient in English; and (d) having lived in North America for at least one year. Participants were recruited through community organizations, churches, email outreach, and word of mouth. In total, 25 participants volunteered for the study.

Although participants were recruited from two different North American countries—the United States and Canada—we approached the analysis collectively, as both countries share significant socio-cultural similarities [18]. Both are Western immigrant-receiving nations with comparable values around individualism, gender roles, and family dynamics, which shape how marital issues are experienced and interpreted. These two countries are also preferred immigration destinations for Nigerians in NA [19,20].

### Data collection and instrument

Participants were screened using a checklist based on inclusion criteria. After obtaining written consent via email, the data was collected through a semi-structured interview between July 2021 and September 2021. The duration of each interview was 60–90 minutes, conducted via the audio Zoom platform, with participants using pseudonyms to protect their identities. The choice of platform was due to the COVID-19 pandemic shutdown and participants living in several US states and Canadian provinces.

All interviews were conducted by trained female researchers with prior experience working with immigrant populations. Although gender differences between interviewers and some participants may have been present, participants were made to feel comfortable and respected throughout the process, and no concerns were expressed regarding interviewer gender. Interviews were conducted via Zoom without video to preserve participant anonymity and support open dialogue. Each session took place in the privacy of the participant's home at a time they selected to ensure confidentiality and comfort. Participants were informed they could pause or stop the interview at any time. At the end of each session, participants were informed about the researchers' availability to provide mental health referrals and counseling as needed. One of the researchers is a mental health provider.

The semi-structured interview guide was developed based on a literature review, the research team's prior study results [16], and consultations with experts in the field. , The interview guide included questions on the prevalence of marital dissatisfaction within this population and what the participants perceived as contributing factors to marital dissatisfaction. Suppose the participants did not include key factors from the literature review. In that case, the interviewer may key participants into these factors to know whether they perceive them to be relevant. The questions were framed to be open-ended so that participants could freely discuss their perceptions. The interview guide covered key areas such as cultural issues, financial instability, domestic violence, and gender role conflicts. Interviews were audio-recorded with participants' consent and transcribed verbatim for analysis. If the participant is married, only one spouse was interviewed in privacy. Although 25 volunteers were available, we conducted and analyzed interviews concurrently, assessing saturation after every three transcripts. Saturation was reached by the 12th interview, as no new concepts emerged beyond that point. However, to ensure confirmability and depth, data collection continued until 15 participants were interviewed. A total of 15 participants completed the interview process.

## Data analysis

This study utilized a descriptive phenomenological design to examine NINA's perceptions and experiences regarding factors contributing to marital dissatisfaction. Rooted in the philosophical work of Husserl, phenomenology emphasizes exploring the way phenomena are experienced as they appear to individuals [21]. Descriptive phenomenology requires researchers to adopt a phenomenological attitude by bracketing personal biases, preconceptions, and natural attitudes to remain fully open to participants' lived experiences [22]. The researchers practiced bracketing through self-reflection, journaling, and revisiting bracketing during data collection and analysis. . . .

Descriptive phenomenology is particularly suited for eliciting detailed descriptions from participants and uncovering the essential elements of a phenomenon as experienced by different individuals. This methodology was deemed ideal for this study because of the limited prior research. It facilitates a deeper understanding of poorly understood phenomena by focusing on the distinct and shared features of participants' experiences.

The study adhered to Giorgi's data analysis framework, an approach inspired by Husserl, which adds rigor and transparency to phenomenological research [22]. This framework ensures that the findings faithfully represent participants' descriptions, preserving their intentional meanings without alteration. This approach aligns with the study's goal of remaining closely connected to participants' lived experiences [22]. Qualitative data from various sources, including field notes, memos, observations, and interviews, were used for the analysis. Interview recordings were transcribed verbatim for detailed analysis. The collected data was then compared and analyzed to identify emerging patterns using the NVivo software. All three researchers independently analyzed the data before comparing their interpretations. Conducting separate analyses first enhanced trustworthiness, credibility, and rigor by minimizing individual bias, ensuring triangulation, strengthening confidence in the findings, and promoting reflexivity [23].. . .

The initial phase involved reading and reading to identify significant statements. In NVivo, we systematically organized and categorized the data to identify key themes using imported transcripts and field notes. Through initial (open) coding, we highlighted relevant text segments and assigned descriptive labels (nodes) based on emerging patterns. Next, axial coding was used to group related codes into broader categories by refining connections between concepts. Finally, we conducted thematic analysis, identifying overarching themes to ensure they accurately represented participants' experiences.

The results were validated against the original data to ensure accuracy and proper reflection of the participants' experiences. Member-checking and peer debriefing between the three researchers were used to validate our interpretations. Results were presented in a narrative format, and quotes were selected to validate the findings.

Credibility was established by maintaining extended engagement with the data and providing detailed, comprehensive descriptions. To ensure the quality and reliability of the data, the researchers implemented strategies such as creating an

audit trail and engaging in reflexive journaling. The researchers utilized the audit trail to uphold data integrity, meticulously documenting data through memos and coding.

### Ethical considerations

Ethical approval was obtained from the Andrews University Institutional Review Board (IRB protocol # 21-071). After explaining the study and clarifying participants' questions, written informed consent was secured from all participants. Participants were assured of confidentiality and the right to withdraw from the study at any time without consequence. Confidentiality was maintained with the use of pseudonyms.

All Zoom interviews were conducted using the university's licensed Zoom accounts, which comply with institutional data privacy policies and are protected by end-to-end encryption. Interviews were audio-only, with no video or screen names used, to enhance anonymity. Recordings were saved locally to secure, encrypted university drives and not stored on the Zoom cloud. Pseudonyms were used throughout data handling and reporting. All electronic files were password-protected, encrypted, and accessible only to the approved research team.

## Results and discussion

Table 1 shows fifteen participants' specific demographic information. Most participants are married (93. 3%), and most reside in the USA (60.0%). Most participants are females (66.7%) and are married in Nigeria (73.3%). Table 2 shows the mean age of the participants, years of marriage, and how long they have lived in NA.

The study's participants were predominantly middle-aged Nigerian immigrants, with most aged between 31 and 60. The majority were female, married, and living with their spouse. Obtaining both male and female perspectives offered a balanced perspective with the current experiences of spouses living together. Many participants were married in Nigeria before migrating to North America, reflecting a strong link between pre-migration marital relationships and their experiences in a new setting. Most participants resided in the US, while a smaller group lived in Canada. Although participants lived in two countries, their experiences were similar, which enabled a detailed description of NINA's experiences in both countries.

We asked participants this question: "How common do you think marital dissatisfaction among NINAs is?" Nine out of 15 reported that marital dissatisfaction is common among Nigerian immigrants to NA. The other 6 participants indirectly alluded to it but did not commit to it being common. One of the participants alluded to the fact that married couples may have marital dissatisfaction before immigrating to NA; however, they feel emboldened to divorce or separate in NA.

Ten themes and eight subthemes were derived from the analysis, as shown in Table 3. The study identified several perceived factors causing marital dissatisfaction among NINAs. These factors include cultural conflicts, particularly around gender roles, creating tension as traditional patriarchal values clash with North American norms of gender equality. Financial stress, stemming from unemployment, underemployment, and the obligation to support extended family in Nigeria, adds pressure to marital relationships. Social isolation due to the lack of familial support further exacerbates these challenges, while acculturation stress from adapting to a new culture causes misunderstandings between spouses. Abuse, both emotional and physical, and infidelity are also highlighted as major causes of marital dissatisfaction, with immigration-related issues like visa status differences contributing to marital instability. The study also found that a lack of essential marital relationship skills—such as communication, respect, and sexual intimacy—left NINAs ill-equipped to handle the stress of emigration and adaptation, exacerbating marital dissatisfaction. The following sections present and discuss the results for each theme.

### Cultural factors

All participants (15) report cultural issues as impacting the relationships between Nigerian immigrant spouses. They observed that the cultural differences between NA and Nigeria often lead to misunderstandings and culture shock as both

**Table 1. Participants Demographics.**

| Variables | Category | Number (%) |
|---|---|---|
| Age | 31-40 | 5(33.3%) |
| | 41-50 | 4(26.7%) |
| | 51-60 | 3(20.0%) |
| | ≥61 | 3(20.0%) |
| Gender | Male | 5(33.3%) |
| | Female | 10(66.7%) |
| Marital Status | Married (living with spouse) | 14 (93.3) |
| | Married (living together but emotionally separated) | 0 (0%) |
| | Married (Living together but physically separated-for example, in different rooms) | 0 (0%) |
| | Legally Separated | 0 (0%) |
| | Widowed/er | 1 (6.7%) |
| | Divorced | 0 (0%) |
| Where Married | Nigeria | 11 (73.3%) |
| | USA | 3 (20.0%) |
| | Canada | 1 (6.7%) |
| If currently married: Years of marriage | 1-10 | 5 (35.7%) |
| | 11-20 | 3 (21.4%) |
| | 21-30 | 3 (21.4%) |
| | ≥31 | 3 (21.4% |
| Country/State/Province of Residence | Canada (2 provinces) | 6 (40.0%) |
| | USA (7 states) | 9 (60.0%) |
| How long have you lived in North America? | 1-10 | 7 (46.7%) |
| | 11-20 | 3 (20.0%) |
| | 21-30 | 4 (26.7%) |
| | ≥31 | 1 (6.7%) |

**Table 2. Mean and Standard Deviation of Demographic Variables.**

| Variable | Mean ± Standard Deviation |
|---|---|
| Age (years) | 47.80 ± 10.73 |
| If currently married: Years of marriage | 19.71 ± 12.99 |
| How long have you lived in North America? | 15.33 ± 12.26 |

spouses adjust to their new environment. Perceived cultural factors identified by participants as impacting marital dissatisfaction are gender issues, extended family issues, acculturation, cultural mindset, and inadequate social support. These factors are described below:

## Gender issues

Fourteen of the 15 participants identified Nigeria's patriarchal culture as a major source of conflict for immigrant couples. They described Nigerian men as dominant in relationships and women as subservient, regardless of their educational status. These participants noted that women were often excluded from family decision-making. Additionally, they observed that many Nigerian immigrant couples adhere to patriarchal cultural practices without questioning them, as these norms

**Table 3. Identified themes and subthemes.**

| Main Themes | Number of Participants Identifying the Themes (%) | Sub Themes | Number of Participants Identifying the Themes (%) |
|---|---|---|---|
| Cultural Factors | 15 (100%) | Gender Issues | 14 (93.33%) |
| | | Extended Family Issues | 12 (80%) |
| | | Acculturation | 6 (40%) |
| | | Cultural Mindset | 5 (33.33%) |
| | | Inadequate Social Support | 3 (20%) |
| Job Situations and Finances | 15 (100%) | Financial Problems | 14 (93.33%) |
| | | Overwork | 7 (46.67%) |
| | | Career Choice | 5 (6.67%) |
| Suffering In Silence | 14 (93.33%) | | |
| Abuse Factors | 12 (80%) | | |
| Lack of Relationship Skills | 11 (73.33%) | | |
| Third-Party Influence | 8 (53.33%) | | |
| Infidelity | 7 (46.67%) | | |
| Parenting Conflict and Childcare Stress | 5 (33.33%) | | |
| Immigration Factors | 4 (26.67%) | | |
| Religion/Spirituality | 3 (20%) | | |

are deeply ingrained from their upbringing, leading to marital dissatisfaction and cultural conflicts. Participants explained that in Nigeria, men have significant autonomy in marriage without being questioned by their wives. However, in NA, women are more likely to challenge male authority, which can intimidate Nigerian men accustomed to traditional gender roles. This resistance from women often creates tension, which they believed contributed to marital dissatisfaction.

Eight participants observed that most Nigerian boys are culturally raised to avoid household chores due to the traditional view of men as providers and women as homemakers. As a result, an immigrant Nigerian woman without house help in NA may struggle with all the household duties, leading to marital dissatisfaction or silent suffering. Participants reported that in Nigerian patriarchal culture, men typically have the final say in most family matters. However, in the diaspora, a woman's earning power can shift decision-making dynamics, potentially challenging the man's traditional role and wounding his cultural ego. Additionally, participants observed that because Nigerian men are often not raised to participate in household chores, the responsibility of managing both work and home life falls heavily on women. This added burden frequently leads to stress and marital dissatisfaction.

*During upbringing, the man understands it is a woman's responsibility to do the chores. As a man, I understand my wife should wash my clothes, serve food, go to the markets, and care for the children. That is the upbringing of some men when they grow old and get married, so they find it difficult to see and appreciate anything different. P 12*

*We hear stories about nurses, and once they make more money than men, they feel that the woman will start bossing them around, or the woman wants to be the man because she earns more, all the ego fights will start. All because the woman makes more money. The men feel threatened; they are not happy that this is happening. P 11*

Fourteen participants reported that Nigerian women's traditional role of childcare and household management is often undervalued by their spouses, leading to dissatisfaction among women. For Nigerian immigrant women in NA, this lack of appreciation is particularly frustrating, as they seek recognition as productive and valued individuals, similar to other women in their new environment. Participants noted that North America fosters a more supportive atmosphere for

women's empowerment, which starkly contrasts with Nigerian patriarchal norms. They observed that in North America, women have greater autonomy, including the ability to divorce or remove a husband from their home. This differs significantly from Nigerian patriarchal culture, where men often wield unchecked authority, sometimes leading to abuse with little or no consequence.

Fourteen participants observed a role reversal in North America, where women increasingly take on the role of breadwinners and expect their husbands to share in household responsibilities. They perceived that this role reversal left many Nigerian men feeling powerless. This shift in power dynamics, fueled by women's employment and empowerment, often leads to feelings of disrespect and dissatisfaction among Nigerian men, creating conflicts and attempts to force women into submission. For women who do not fully embrace this newfound freedom and empowerment, their husbands may continue to expect a subservient relationship, leading to suffering in silence for the women who resist this cultural shift.

*In Nigeria, people think twice before they want to separate: what will people say? Oh, what will my mother say …. but here, people do not think about what people say, or you just go to court, walk into the court, or send a paper that you want to be separated. So, I think there is more freedom or liberty to do what you want here, compared to Nigeria, because of the laws. P 5*

*The expectations that go with living in another cultural environment. This affects role expectations and who should be doing what; we work and do not have help. In Nigeria, the roles and responsibilities are not so blurred. The woman keeps the house; even though she goes to work, she still runs the house in Nigeria. Most of the time, the man believes his job is to provide for the family as long as he brings the money in. This should be cool, um, so when they get here; it is not like that – the men are supposed to be involved in also running the house and also doing school work, taking care of the kids, so they have a cultural shock. Most of them don't prepare for that, so when they come, it is a major area of dissatisfaction where the woman feels you can't leave everything to me. P 12*

## Extended family issues

Twelve participants identified the demands of the extended family as a contributing factor to the cultural issues experienced after immigration to NA. Participants reported that Nigerian culture emphasizes financial dependency on "successful" immigrant families, placing pressure on Nigerians in North America to support relatives in Nigeria, often at the expense of their own nuclear family's financial well-being. In this context, the extended family includes in-laws, distant relatives, and even individuals who consider themselves family. Additionally, participants observed that Nigerian cultural norms dictate that children are responsible for caring for their parents in old age and are expected to obey them, even in adulthood. This expectation of NINA caring for extended family members is very stressful for young NINA couples, especially when they invite two in-laws into their NA home simultaneously. Participants reported that when in-laws visit or move to the U.S. to live with their children, their presence can create tension in marriages, particularly when husbands prioritize their parents' opinions over their wives in decision-making. Some in-laws exert significant control over Nigerian immigrant marriages by reinforcing cultural expectations, including traditional gender roles, financial obligations, and relationship dynamics, which often lead to marital dissatisfaction. Additionally, participants noted that when in-laws disapprove of a spouse, it further intensifies marital strain and dissatisfaction.

*The mother in-law expects her son to come from work, cross his legs while watching TV, and then get served with everybody. You know it doesn't work that way here (NA). They see their grandchildren putting clothes in the laundry, ironing, or washing plates. Oh, My Goodness, 'won ti fi iya je. won ti fi ori e gba paro'(He is being tortured. They have exchanged his destiny) P9*

*In our culture, man is expected to provide for the extended family. Men are expected to care for their mom and dad when they reach a certain age, which is okay. It is accepted. It is an expectation all around, regardless of whether we live here or will live over there. However, it is the other people like aunties and uncles and brothers and sisters, and you have got to pay their school fees. That is when it becomes exhausting for some of these young couples. P 10*

## Acculturation

Six participants identified assimilation into American culture as a source of stress in marital relationships. They observed that exposure to American values, particularly those emphasizing individual freedom, can lead to a shift in cultural boundaries, where couples feel free to act without the traditional constraints of Nigerian norms. This shift sometimes creates tension when one spouse adopts aspects of American culture while the other maintains traditional Nigerian expectations. For example, participants noted that Hollywood influences introduce behaviors such as men opening car doors for their spouses—an act uncommon in Nigerian culture. When one partner embraces such behaviors and the other does not, it can lead to misunderstandings and marital dissatisfaction.

*More and more people were coming to realize or just pattern their marriages after maybe what they see at home. Then, when they enter a different culture where marriage and marital roles are a bit different, it throws everyone off. P4*

## Cultural mindset

Five participants highlighted deeply ingrained cultural mindsets that conflict with North American norms, creating marital conflicts and dissatisfaction. These mindsets include (a) financial obligations to extended family in Nigeria, sometimes prioritized over the nuclear family's needs; (b) the belief that, after a divorce, the woman should leave the home rather than the man; (c) the assumption of male dominance in relationships, particularly in financial decision-making; and (d) the expectation that parents continue to provide guidance and influence over their married children's relationships. Participants perceived that these cultural values, deeply rooted in their upbringing, often clash with North American societal norms, leading to internal and relational conflicts that persist long after immigration, causing marital dissatisfaction.

*However, that's not the mentality in Nigeria; no matter how horrible the man is, it is the woman who will go at the end of the day (after a divorce). P9*

## Inadequate social support

Three participants highlighted the stark contrast in support systems available for families, particularly concerning childcare, between Nigeria and NA. They reported that in Nigeria, extended family and community networks provide substantial help with childcare and marital mediation. In North America, it was observed that many feel isolated in parenting and resolving marital issues, with support typically limited to the few who have close family members, such as parents, living nearby to assist with childcare. This lack of support creates additional stress on the marriage relationship.

*The major difference is the help you get with the kids in Nigeria; you have family and friends surrounding you. Here, you are on your own except for the lucky few who have their parents with them and can help with the kids. P8*

## Discussion on cultural factors

Nigerian Immigrants in North America (NINAs) face cultural conflicts that contribute to marital dissatisfaction. This issue is not unique to NINAs; Sim et al. reported similar cultural adaptation challenges among Mexican immigrant couples [24].
    Another source of marital dissatisfaction that was identified in our study was differing cultural expectations after divorce. In Nigeria, it is customary for women to leave the matrimonial home, forfeiting property rights and entitlements [25]. In

contrast, North American laws often allow women to remain in the home and ensure equitable asset division. This role reversal can cause frustration for Nigerian men, sometimes discouraging them from pursuing divorce and escalating marital tensions, potentially leading to intimate partner violence (IPV). Meanwhile, Nigerian women in North America may feel more empowered to express dissatisfaction due to legal protections that favor gender equity.

Cultural obligations, such as adult children supporting aging parents, also strain NINA marriages. Balancing care for parents, either locally or through financial support, often competes with marital priorities, increasing dissatisfaction.

Furthermore, patriarchal expectations from Nigerian culture often clash with North American norms. Men may expect wives to manage domestic duties regardless of their professional status, while North American values emphasize shared responsibilities and gender equality. When women become primary earners, this role reversal can challenge traditional dynamics and contribute to marital strain.

### Job situations and finances

The job and financial issues identified by all 15 participants were derived from three subthemes of financial pressures, overwork, and career choices, as described below:

### Financial problems

Fourteen participants identified financial pressure as a significant factor contributing to marital dissatisfaction among NINAs. Participants observe these pressures to mainly affect Nigerian men's sense of self-worth, often leading to the need to overwork in order to earn more money. They reported that in many cases, Nigerian women in North America are more likely to secure better-paying jobs than men, which can result in jealousy and envy within the marriage. Participants noted that some Nigerian men struggle to find employment that matches their educational qualifications and may refuse lower-status or menial jobs, choosing instead to remain unemployed. Even with legal residency, they are perceived to have only moderate chances of securing well-paid positions. As a result, some men seek employment in distant locations to earn a better income, leading to prolonged physical separation and added strain on the marriage.

*So, when people come to Canada when they come in and they cannot get the kind of jobs that they want, they are underemployed. Oh, that makes them unhappy in their career and makes them cranky, unteachable, and all of that. Then, it translates into the marital relationship, such as if the man is under-employed and is not happy with the kind of job he is doing. The woman happens to be having a better time now in her career, you now see jealousy, you see envy. It should have been me. Why is it not me who is going to pay the mortgage? The mortgage was not in my name, etc. P 11*

The 14 participants also reported that when Nigerian women earn more than their husbands, they often expect to have a say in family financial decisions, leading to conflicts. In traditional Nigerian homes, participants observed that men are typically the primary breadwinners. The shift in financial dynamics after immigration was reported to damage men's ego, contributing to marital dissatisfaction. In response, some men were observed to project an image of wealth. To maintain this façade despite lower income, they often work long hours. However, under economic stress, this illusion tends to collapse, increasing marital dissatisfaction.

*The men are not honest with themselves. They want to project a certain image while suffering in the background. This comes with the mindset that you must outdo the next person. P 10*

Another contributor to marital dissatisfaction reported by eight participants was the lack of joint financial accounts among Nigerian couples, which led to financial disagreements and a lack of transparency. This secrecy around finances was observed to often result in conflicts over spending and saving. The participants highlighted that NINA experience additional

financial pressures which include paying bills, which is uncommon in Nigeria, and the obligation to support extended family members in Nigeria. They opined that tensions may arise if the husband takes money from his wife to send to his extended family or reckless spending by either spouse, particularly by some Nigerian women, further strains the marital relationship.

*In Nigeria, when you build a house, it is your house, the car is your car, the land is your land, things you know you do not have to pay monthly or mortgage. That is what I meant, so there is more financial responsibility here. …….So that is what I meant by saying that there is more responsibility for bills that come in every month in NA than in Nigeria. P 5*

*As a low-income family who had just come to NA, the husband was traveling home (Nigeria), and the money he spent at home was outrageous. He bought the latest model of one car in the range of 6 million Naira. Moreover, I was like, for what? Who are you buying this for? What are you doing this for? Why would you do a thing like that? Why would you spend this kind of money to get this car? Just for show? At the end of the day, you may not even sell it. P 12*

### Overwork

Seven of the 15 participants highlighted that many Nigerians in North America are often driven by a desire for success and affluent living and are influenced by their peers. This aspiration was observed to frequently lead to overworking to maintain a lavish lifestyle, which in turn negatively affects the time invested in marital relationships, particularly in terms of emotional and sexual intimacy. According to the participants, this pressure to "live large" like others is a common challenge among NINAs. It was reported that both partners are often overwhelmed by work demands, creating stress at home. This strain was noted to be acute for women, whom the participants identified as being typically responsible for most of the childcare and household duties along with their jobs, further exacerbating dissatisfaction within the marriage.

*I think both the man and the woman and the experience is pretty much the same. I work, and the man also works. The woman feels she has been left alone to care for the kids. Of course, that plays out differently here because what that means is she drops them off at school, then right after school helps them with homework and runs them to other activities—caring for them in terms of feeding, clothing, and ensuring pretty much that her kids do not look crazy. When they go out there, she has her job on the side, and she has to show pretty much at work that everything is okay. Of course, by the time the husband gets home, the wife is exhausted, so she cannot even function or carry out her marital duties as effectively or as much as the husband wants her to. P4*

*Because Nigerians like good things, you know we want to live a lavish life. You know we want to make it; we are very curious, and in North America, there are so many opportunities given to us to excel in terms of career compared to Nigeria. So anyone who comes here and sees this opportunity wants to grab it; they want to jump into it and make what they can out of it. Not realizing it is taking them away from their ability to satisfy their spouse as husband and wife. P5*

### Career choices

Five participants indicated that residing in North America facilitates financial independence, particularly in healthcare professions such as nursing. They noted that Nigerian men generally support their wives' nursing education in this region. However, participants observed a shift in wives' attitudes post-training, as financial self-sufficiency challenges traditional Nigerian marital roles and raises concerns about the initial willingness to share financial responsibilities. Additionally, they highlighted the nursing profession as both a highly sought-after and vulnerable profession among Nigerian Immigrants in North America (NINAs).

*I mean, there are, especially nurses from Nigeria, the women. Most of the time, people say nursing – that is where the money is. While in the Nursing program, their husbands took care of the bills and everything else, ensuring everything*

*was going well. By the time they finish their nursing and then pass their exam, automatically, the man will feel that at least now they are making money, so at least they can be part of the financial solution of the family. P 2*

## Discussion on job situations and finances

The participants' accounts shed light on the significant impact that career choices, financial pressures, and overwork have on the marital satisfaction of NINAs. One prominent subtheme is career choice, particularly in healthcare, where Nigerian men often support their wives through nursing training. However, the shift in financial dynamics post-training, with women gaining financial independence, challenges traditional Nigerian marital roles. This financial role shift has been noted in the literature, where changing economic power within marriages often leads to conflicts as immigrant women, especially nurses, become more financially autonomous [26]. Financial stress, particularly among Nigerian men who face underemployment or unemployment, often creates economic imbalances between spouses, leading to feelings of inadequacy and jealousy. This inequality can lead to financial secrecy, as joint accounts are uncommon in Nigerian culture, where men are traditionally the providers, and financial burdens are lighter since most items in Nigeria are paid for upfront rather than through ongoing bills. After emigration, disparities in earning power and disagreements over spending from joint accounts increase tension, especially when there is no consensus on financial decisions. Additionally, the obligation to support extended family in Nigeria exacerbates conflicts, mainly if one spouse's family receives more financial support than the other.

Another finding is that the desire to "live large" among NINAs impacts on marital satisfaction by driving both partners to overwork. Many feel intense Nigerian societal pressure to attain a high standard of living, influenced by the success and lifestyles of those around them. This drive for success and a "lavish life" can also shift focus away from maintaining a strong marital relationship, as observed in Participant P5's experience. This ambition also leads to both partners working long hours, sometimes at multiple jobs, to achieve these goals. This aspiration to "live large" creates a cycle where financial goals are prioritized over emotional and relational needs within the marriage. Long hours and high work demand frequently lead to burnout, reducing time and energy for family life. Gopalan et al. found that work-family conflicts, especially among immigrants, often decrease marital satisfaction by creating stress and emotional distance between partners [27].

Nigerian women face the added challenge of balancing professional roles with household duties, leading to high levels of exhaustion. Ajala highlights how the "second shift" of managing work and home affects women's ability to fully engage in their relationships, leading to marital dissatisfaction when both partners feel the impact [28]. Balancing roles aligns with the experiences of participant P4, who noted that, after a long day of balancing work and parenting tasks, many women feel too exhausted to meet their partners' expectations for intimacy. The cultural emphasis on financial success also adds to this strain. Many Nigerian immigrants prioritize economic achievement, sometimes placing it above family life, further exacerbating marital dissatisfaction [29]. Molina's study highlights how work-family conflict, intensified by a focus on career success, negatively impacts personal relationships, creating a need for balance to maintain family harmony. Hines discusses how immigrants pursuing the "American Dream" often isolate themselves from the family support systems they once relied on [30]. This isolation exacerbates the mental and emotional strain of overworking and further challenges marital satisfaction [31].

## Suffering in silence

The "suffering in silence" theme emerged in 14 participants' responses. They reported that Nigerian women in North America often endure hardship in their marriages without seeking help, as Nigerian societal norms discourage them from discussing abuse or marital issues publicly. The fear of stigma and concerns about the legal consequences their husbands may face in North America was perceived as preventing many women from "airing their dirty linen in public." All 14 participants believed that therapists and counselors in North America, unfamiliar with Nigerian cultural values, tend to

recommend divorce or separation, which conflicts with Nigerian traditions that emphasize preserving the marriage and enduring suffering for the sake of the children and societal expectations.

The 14 participants also noted the absence of the elders, kinsmen, and extended family members, who traditionally intervene and mediate marital conflicts in Nigeria. Without these support systems in NA, participants observed that Nigerian women may be more open to counseling, while Nigerian men are generally reluctant to discuss their marital issues.

*With the mindset of Nigerians who come abroad, one of the worst things you could do is to call the police on your spouse. So, some ladies try to maintain peace and do not report some of these abuses. P3*

*In Canada. I am sorry to say this, but from my experience – they (counselors) are not pro-marriage. So, because they are not for marriage the slightest report of the tiniest resemblance of abuse – they are ready to encourage their partners to end the marriage. P11*

*The Nigerian man should not be a 'Sisi.' Because they feel that for somebody else to sit on the other side of the table to talk to them about their marriage is insulting, so I feel like a good number of Nigerian men are not there yet, where they cannot sit with a counselor or therapist, I think it bruises their ego a lot. P4*

**Discussion on 'suffering in silence.'**

In our study, "suffering in silence" describes individuals enduring profound pain or struggle without openly sharing their experiences. This silence often arises due to societal, cultural, economic, and religious pressures, which can cause Nigerian immigrant partners to withhold their struggles, even when facing mental health challenges. Traditional expectations for Nigerian women emphasize submission and preserving family harmony, regardless of personal hardships, with marital issues typically resolved privately, away from public scrutiny [31]. Speaking out against domestic abuse, violence, or infidelity is often viewed as a sign of weakness, disloyalty, or failure [31].

Religious beliefs also play a significant role in Nigerian marriages as a silencing factor, where couples are encouraged to pray rather than seek external help during marital conflicts. This approach can lead to prolonged silence as individuals continue praying for a resolution, sometimes without tangible results [32]. Additionally, the fear of societal judgment and the involvement of extended family, who may advocate for silence and endurance, contributes to the tendency for one or both partners to suffer in silence [33]. Additional factors, including the hardships endured to shield children from the stigma of a "broken home" and the emotional challenges associated with infidelity or polygamy, are emphasized in Siddiq's exploration of women's struggles. These issues contribute to why Nigerian couples may 'suffer in silence.' [34].

**Abuse factors**

Twelve participants described abuse as a perceived significant cause of marital dissatisfaction among NINA. The abuse was described as physical, emotional, and verbal, sometimes leading to homicide. The causes of abuse described include financial hardships, stress from work, men's low self-esteem, and women's financial independence. Participants perceived abuse as common among nurses. NINA men usually encourage their wives to be nurses in NA because of their professional earning power. This imbalance in earning power was observed to lead to financial intimidation and abuse from the man. Participants reported they perceived abuse within Nigerian immigrant spouses is usually unreported because of fear of retaliation, the effect on immigration status, and the unknown about law enforcement's response to the report. They described this phenomenon as 'suffering in silence' which may lead to homicide. For example, a participant observed the following:

*I remember a case in Canada where a man stabbed his wife. I heard the man saying he first shot the woman in the face, and the woman did not die, but the ambulance came, and he said no, I am not going to stop. I am going to kill*

*her and kill myself. He said that since she became a nurse, she did not respect me and would not allow me to do my prayers or give my family money.* P1

The twelve participants also described the perceived consequences of abuse, highlighting its impact on the extended family. They noted the emotional and physical toll on children, which may contribute to a cycle of abuse in future generations. According to the participants, the abuse may start before the couple emigrates; however, it is worsened by the stress of a new environment and all its challenges.

*Some Nigerians, growing up, have seen their fathers get scot-free doing it. Many kids have seen their mothers go through this, not just physical abuse but emotional abuse, too. Which is sorry to say – emotional abuse is worse than physical abuse. For example,*

*calling a woman names about their body, "o kan tobi" (You are just fat for nothing).* P8

### Discussion on abuse factors

Marital dissatisfaction among NINAs often stems from instances of spousal abuse, which can manifest as physical, emotional, or verbal aggression. In severe cases, these abuses may escalate to tragic outcomes, including homicide [35]. The stressors associated with immigration, as stated in our study, such as cultural differences, financial pressures, and overwork, sometimes lead individuals to project their frustrations onto their partners, resulting in abusive behaviors. Studies suggest that limited coping mechanisms and poor stress management exacerbate these conflicts within immigrant Nigerian marriages [36].

### Lack of relationship skills

Eleven participants perceived a lack of relationship skills as a major contributor to marital dissatisfaction among NINAs. Deficiencies in empathy, communication, prioritization, and respect were highlighted, with many couples focusing on immigration issues, work, childcare, and education rather than nurturing their marital relationship. Participants also observed the inability to balance these responsibilities often led to emotional disconnection. Inadequate premarital counseling, particularly the insufficient guidance provided by Nigerian religious leaders, was perceived to leave many couples unprepared for the realities of marriage. Sexual relationship skills were another area of concern observed, as couples struggled to navigate differences in preferences and boundaries. The added strain of migration impacted household responsibilities and financial instability was observed to disrupt intimacy. This lack of preparation and ongoing marital challenges were reported to ultimately lead to marital dissatisfaction for many NINAs as they navigate the foreign terrains of NA.

*The culture in itself is that in our culture, we do not prepare our children for a union. We prepare them for a wedding, not a union.*

*Moreover, when I say that it is not for a union, traditionally, we encourage the kids to get educated, which is good. Then, the next step is to encourage them to find a life partner and just be married without really taking time to educate them on what it means to be committed to somebody for life.* P 10

### Discussion on lack of relationship skills

NINAs were found to lack some key marital relationship skills, which could mitigate the challenges of migrating to North America. Such relationship skill deficit is a lack of respect for unemployed husbands when wives are the primary earners

or earn more than their husbands, thus challenging traditional gender roles. Research by Akanle et al. indicates that wives who become breadwinners may show less affection, leading to marital dissatisfaction [33]. Additionally, many Nigerian couples are unprepared for the unique marital challenges they encounter in North America due to inadequate premarital counseling, which is often conducted in Nigeria by religious leaders who may lack formal training in professional marital counseling. Furthermore, exposure to diverse sexual preferences in NA, as opposed to predominant conservative sexual practices in Nigeria, may cause marital dissatisfaction among NINAs when couples differ in their sexual expectations and desires. Alabi and Olonade attest that historically, Nigerian couples hold conservative beliefs about sex and sexual behaviors that are challenged by modernity and exposure to first-world mindsets [37].

### Third-party influence

Eight participants observed that third-party influence from peers and movies can negatively cause marital dissatisfaction. Participants described how the drive for competition to "keep up with the Joneses" influenced women to spend more money and compare their husbands to other men, expecting them to provide everything needed for the competition.

> He could not buy a house yet, and this young lady had friends discussing their houses, backyard, and swimming pool. Nevertheless, she could not discuss it at that level with them, and she was unhappy, so you (the husband) must do something; you must get a house. P 6

### Discussion on third-party influence

Nigerian immigrants often try to emulate their peers, both within and outside their cultural community, particularly regarding the kind of homes they live in, the cars they drive, and the lifestyle they lead, including popular vacation destinations. To keep up with these standards, some Nigerian immigrants work excessively to afford this lifestyle, striving to "keep up with the Joneses." The influence of Hollywood portrayals of relationships was also noted, as these often contrast with Nigerian cultural expectations, sometimes resulting in marital conflicts. Arthur describes this tension as a clash between the realities of American life and the immigrant's desire to maintain an African cultural identity while pursuing the American dream [38].

### Infidelity

Seven participants stated that polygamy, which is permitted by the Nigerian culture but prohibited by NA laws, is an excuse for men but not women to engage in infidelity. It was observed that couples engage in having multiple partners even when separated but are still living together under the same roof. Lack of sexual satisfaction/attention, deprivation, tiredness, long-distance relationships due to work, and emotional disconnections were also identified as causes of infidelity.

> Culture again gives excuses for men when they cheat on their wives from back in the days when a man does it just like ah, you know," Okunrin lo'nse" (He is a man and men do this), you have to forgive him and all that nonsense. However, when a woman does it, there is no forgiving there. P 8

### Discussion on infidelity

This study highlights how cultural norms, such as the acceptance of polygamy in Nigerian culture, intersect with legal restrictions in North America to influence perceptions of infidelity among Nigerian immigrants. In Nigeria, customary laws allowed for polygamy [39]. After immigration to the USA, because of the restrictive laws with polygamy, some Nigerian

men continue with extramarital affairs, causing spousal dissatisfaction. These cultural norms align with recent studies that indicate how cultural beliefs about polygamy can persist after migration, affecting marital dynamics [40,41]. Participants highlighted that emotional separation, where spouses remain under the same roof due to financial constraints or fear of judgment, allows infidelity while maintaining the appearance of a stable marriage. This emotional disconnection often leads to sexual dissatisfaction and further strains relationships.

## Parenting conflict and childcare stress

Five participants observed that differing child-rearing practices between Nigeria and NA and the challenges women face in managing childcare responsibilities as contributing to marital dissatisfaction. The effect of the father being absent from home due to overwork, constant quarreling, and misunderstanding among spouses was reported to have a toll on the children in the family. Also, when fathers are unemployed, children seek jobs to help parents, which perpetuates tension and stress for the children.

> *Their 13-year-old son said he needed a job this summer, so he wants to get out of the house and does not want to be at home with Daddy. Daddy does not go to work; his daddy is at home. So, this child wants to go and get a job. If the child wants to work, how many hours will he work daily? You mean, he will make some money, and the children already know there is tension at home.* P11

> *Okay. However, she is so exhausted inside her that she doesn't have time for her kids. I looked at her lifestyle, and both parents are seriously working to meet up with some financial obligations. But then you find out that there is no time for the kids. Every time you see the kids, even the mother – She cries – and says, uhh, I do not know what I have gotten myself into. I am working. I tried to kill myself, and yet I am not even seeing what I am working on, not even seeing it. I feel my kids are just so animalistic. They are not just behaving well.* P 1

## Discussion on parenting conflict and childcare stress

Nigerian couples in North America often experience marital tension over child discipline, particularly regarding corporal punishment. In Nigeria, corporal punishment is culturally accepted and legally permitted, while North American child protection laws generally prohibit it [42]. This legal and cultural divide can lead to conflicts in immigrant families, as one spouse may favor traditional practices. At the same time, the other prefers to follow local norms to avoid legal repercussions, impacting overall marital satisfaction. This is not peculiar to the Nigerian culture alone; Wang et al. observed that there was a bidirectional relationship between parenting stress and marital satisfaction among Chinese couples [43].

## Immigration factors

Four participants discussed how immigration-related issues can create significant stress in marriages. Arranged marriages for immigration purposes, differences in immigration status, and the challenges of regularizing status were identified as key stressors that affect marital satisfaction among NINA couples. These issues often lead to nagging, conflict, and poor communication between spouses. Additionally, participants observed that many individuals do not report domestic violence because they fear jeopardizing their chances of becoming citizens, choosing instead to keep the abuse hidden to protect their immigration goals.

Participants also highlighted that many immigrants are unprepared for the cultural, financial, mental, and spiritual demands of life in NA. Upon arrival, they quickly realize the need for hard work and independence, leading to culture shock. This gap between expectations and reality was identified as often resulting in frustration, depression, and emotional tension within families, affecting relationships with both partners and children.

*The man is not regularized, whereas the wife is regularized. She has been a citizen for many years. She says each time there is a problem, go out and find out from your fellow men how they are doing it. Is that how other men do? See how long you have been here; you are not even a citizen. So, it is a frustrating situation. P 6*

*When the man comes here, because they cannot get an immigrant visa, the man goes into an arranged marriage. Then, the wife coming was a big mistake because he was trying to marry another person so they could get a visa. P15*

*In Canada, violence and abuse are usually not reported. They do not do it in a way that the police will get involved, even when the man is abusing the woman. The woman does not want to report just because when you come to Canada as a permanent resident, you must stay three years straight before becoming a citizen. When Nigerians arrive here, becoming citizens is the goal. So they do not want to do anything to jeopardize that simply because they don't want to give their spouse a police record P 11*

## Discussion on immigration factors

The findings of this study reveal how immigration-related stressors impact marital dissatisfaction among NINAs. Participants identified the need to marry a NA citizen to regularize their status through marriage. Some Nigerians may not disclose their Nigerian marital status and choose to marry someone from NA to the detriment of both Nigerian and NA spouses, causing marital conflicts. These stressors are consistent with more recent studies, which highlight the role of citizenship as impacting spousal choice for immigrants [44]. The dependent spouse's fear of deportation or jeopardizing one's immigration case when the status is yet to be conferred often prevents individuals from reporting domestic violence, leading to the concealment of any abuse to protect their immigration prospects [45]. In addition to legal and immigration challenges, participants described inadequate preparation for immigration, leading to a mismatch of expectations and reality [46]. This experience of unmet expectations resonates with current literature, which describes how immigrants need to assimilate into their host country through positional and progressive steps [47]. The unmet expectations lead to stress that negatively affects immigrants' mental health, often resulting in issues such as anxiety, psychological distress, and difficulties accessing equitable healthcare [48].

## Religion or spirituality

Three participants identified religious beliefs, such as support for polygamy and infidelity, as a key factor of marital dissatisfaction. Also, religious practices such as the power of prayer were reported as a resource for navigating and resolving marital challenges. However, participants expressed that upon emigrating to NA, NINAs realized that the power of prayer was not enough to bring healing in toxic relationships. This contrasts with the religious advice frequently given in Nigeria: praying more without seeking additional intervention. Additionally, participants observed that Nigerian societal practices allow men to practice infidelity based on Islamic belief, and women are not allowed to but instead rely on faith and prayer to endure any emotional suffering.

*Let us start with religion; some Muslim men feel as if they want a second wife, it doesn't matter where they are. I got a second wife because I'm Muslim. My father married three wives and may marry four wives, and I'm a Muslim. This is the man talking to the wife – you knew this before, and I can decide to marry two or three wives is the excuse for infidelity. Muslim religion in the Nigerian context feeds infidelity. P8*

*But when I got here, I found out that when one is in a toxic relationship, you cannot be healed while you are in there, but our religion back in Nigeria will tell you, do not worry, stay there, do more praying. P12*

## Discussion on religion or spirituality

Religiosity and spirituality significantly influence marital satisfaction among NINA, often shaping expectations and responses to conflict. Traditional reliance on prayer as a sole intervention or religious justifications, like polygamy (Qur'an 4:3), can sometimes rationalize infidelity or fail to address deeper issues. Exposure to Western counseling methods instigates a query of the traditional religious posture of reliance on prayer alone, which introduces tensions between couples and aggravates dissatisfaction.

## Strengths and limitations of the study

The strength of this study is its comprehensive exploration of the perceptions of the diverse experiences of Nigerian immigrant communities across North America, incorporating insights from participants based in the USA and Canada. Examining the cultural and contextual factors at play illuminates an often-overlooked issue, effectively addressing a significant gap in understanding this under-researched population. However, it is essential to acknowledge that this focus primarily confines the findings to the Nigerian demographic. Despite the relatively small sample size of 15 subjects, the in-depth nature of the individual interviews provides rich, nuanced insights into the cultural elements driving marital stress within this community. As is typical with qualitative studies, this limited sample size affects the results' generalizability.

Nonetheless, most participants being married and living with their spouses allow for a thorough investigation into marital dissatisfaction, making the findings highly pertinent. However, this focus may need to include valuable perspectives from divorced or separated individuals. Although the participant pool comprised more females than males, including male voices enhances the analysis, offering a more balanced view of marital dynamics. Moreover, most participants between 40 and 60 years of age ensure that the insights shared are grounded in mature perceptions of lived experiences. This blend of factors contributes to the relevance and significance of the findings in understanding the complexities of marital relationships within this demographic.

A potential limitation of this study is the recruitment of participants from two different national contexts—Canada and the United States. While this could introduce variability due to differences in immigration and social systems, our analysis did not reveal any systematic differences in participants' responses that could be attributed to country of residence. Instead, consistent themes emerged across both groups, underscoring the dominant influence of shared cultural background and immigrant identity over national policy differences. Nevertheless, future studies may benefit from examining country-specific nuances more explicitly, particularly where institutional or policy differences may affect marital dynamics.

The participants in this study were heterogeneous in terms of socio-demographic backgrounds, including age, marital duration, and length of stay in North America. While this diversity may limit the ability to generalize findings to specific subpopulations, it also enriched the data by offering a wide range of perspectives on marital dissatisfaction. The consistency of themes across diverse participant profiles suggests shared cultural and experiential factors among Nigerian immigrants, although further subgroup analyses could be beneficial in future research.

## Conclusion

This study examined the multifaceted factors contributing to marital dissatisfaction among Nigerian immigrants in North America. The study findings highlighted the interplay between cultural, financial, social, and relational dynamics. Nigerian traditional gender roles often clash with the egalitarian norms of North America, creating conflicts around household responsibilities, economic decision-making, and power dynamics. Financial stress, driven by male underemployment and the obligation to support extended family, leads to overwork and further exacerbates relationship strains. In contrast, the lack of extended family support networks heightens feelings of isolation and stress.

Additionally, cultural adaptation challenges, including shifts in relationship expectations and communication deficits, amplify marital dissatisfaction. These stressors underscore the need for tailored interventions and support systems to

enhance marital satisfaction and well-being within this population. These study findings provide valuable insights for policymakers, counselors, and community leaders planning to support healthier marital dynamics among immigrant populations, especially Nigerians.

## Acknowledgments

The authors wished to acknowledge the input of Dr, Jean Cadet, Chair and Associate Professor of Public Health at Andrews University and Dr. Joan Aina, Adjunct professor at Andrews University during the original planning of this study. We also acknowledge the help of Alyssa Sussdorf and Ashley Ellis, graduate students from Andrews University, for transcribing and arranging the original transcript for analysis. We are grateful to our fifteen participants for their time and robust input into understanding this phenomenon.

## Author contributions

**Conceptualization:** Jochebed Ade-Oshifogun, Augusta Olaore.

**Data curation:** Jochebed Ade-Oshifogun, Augusta Olaore.

**Formal analysis:** Jochebed Ade-Oshifogun, Augusta Olaore, Kwabena Adu Agyemang.

**Funding acquisition:** Jochebed Ade-Oshifogun.

**Investigation:** Jochebed Ade-Oshifogun, Augusta Olaore.

**Methodology:** Jochebed Ade-Oshifogun, Augusta Olaore.

**Project administration:** Jochebed Ade-Oshifogun, Kwabena Adu Agyemang.

**Resources:** Jochebed Ade-Oshifogun.

**Software:** Kwabena Adu Agyemang.

**Supervision:** Jochebed Ade-Oshifogun.

**Validation:** Jochebed Ade-Oshifogun, Augusta Olaore, Kwabena Adu Agyemang.

**Visualization:** Jochebed Ade-Oshifogun, Augusta Olaore.

**Writing – original draft:** Jochebed Ade-Oshifogun, Augusta Olaore, Kwabena Adu Agyemang.

**Writing – review & editing:** Jochebed Ade-Oshifogun, Augusta Olaore, Kwabena Adu Agyemang.

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
