## [Decision Letter · Decision Letter 0]

10 Apr 2025

PONE-D-25-087182/17/2025

Perceived causes of marital dissatisfaction among Nigerian immigrants in North America: A qualitative studyPLOS ONE

Dear Dr. Ade-Oshifogun,

Thank you for submitting your manuscript to PLOS ONE. After careful consideration, we feel that it has merit but does not fully meet PLOS ONE’s publication criteria as it currently stands. Therefore, we invite you to submit a revised version of the manuscript that addresses the points raised during the review process.

We look forward to receiving your revised manuscript.

Kind regards,

Patrick Ifeanyi Okonta, MBBCh, MPH, FWACS, FMCOG, MD, DRH

Academic Editor

PLOS ONE

2. Please ensure that you include a title page within your main document. You should list all authors and all affiliations as per our author instructions and clearly indicate the corresponding author.

3. Please amend your authorship list in your manuscript file to include author Jochebed Ade-Oshifogun, Augusta Olaore, Kwabena Adu Agyemang,.

Additional Editor Comments (if provided):

Reviewers' comments:

Reviewer's Responses to Questions

**Comments to the Author**

1. Is the manuscript technically sound, and do the data support the conclusions?

Reviewer #1: Yes

Reviewer #2: Yes

2. Has the statistical analysis been performed appropriately and rigorously? 

Reviewer #1: N/A

Reviewer #2: N/A

3. Have the authors made all data underlying the findings in their manuscript fully available?

Reviewer #1: Yes

Reviewer #2: No

4. Is the manuscript presented in an intelligible fashion and written in standard English?

Reviewer #1: Yes

Reviewer #2: Yes

5. Review Comments to the Author

Reviewer #1: The paper clearly articulates the perceived factors responsible for marital instability amongst married Nigerian immigrants in North America (USA and Canada). The authors highlighted current thoughts about marital instability, including existing gaps in the literature. Standard reporting guidelines (COREQ), sound theoretical framework and adequate sample size were used to arrive at the findings.

Marital instability, as perceived by the participants, is because of several socio-cultural and economic factors. These are cultural, gender, extended family, acculturation, cultural mindset, inadequate social support, job situation and finances, suffering in silence and abuse factors, lack of relationship skills, third-party influences, infidelity, parenting conflict and childcare stress, immigration issues, and religion or spirituality.

These above key findings are crucial for national migration and socio-welfare policy decisions, not to discourage migrations of married couples or Nigerians seeking greener pastures abroad, but for education and counseling purposes for newly settled couples, intending couples in Nigeria who are planning to migrate or those experiencing marital satisfaction. Although the generalizability of the findings is limited to Nigerians, the insights gained from the interviews could apply to any migrating couple from any part of the world to a foreign land.

Issues for consideration in the revision and for further studies:

1. This research recruited participants from two countries with dissimilar socio-cultural and immigration practice. Could these contextual factors bias the results? This study could be further strengthened by highlighting the nuances in the two socio-cultural milieu in which the participants live.

2. The socio-demographic variables show a heterogenous population. It is important to note this as a potential source of bias in the reporting.

3. More so, some participants, in addition to their different socio-cultural and economic factors, presented with different marital problems. It is expected that participants with troubled relationships would normally not open up in a counseling session, not to talk of during an interview. Although not clear from the researcher’s profile information, there seems to be a minor gender difference between the participants and the researchers. It is interesting how the researchers handled the sensitive nature of the study. Did the researchers interview participants of the same gender? What were the gender considerations? How did the researchers ensure that the information was given in a relaxed atmosphere? These types of participants should be interviewed by trained personnel and, thereafter, for those in need of help, offered some social support after the research. How these were addressed by the authors is important.

4. The age at which couples experience marital instability may not be their early years, at least from empirical and anecdotal reports. So, it is surprising to see that the participants were recruited from a seemingly young (fist year) marital age. Could the findings have shown a different view if those with more years of marriage were interviewed?

5. Using audio instead of a video zoom interview may worsen the inherent challenges of maintaining engagement, observing non-verbal cues and other technological issues. How these were considered in the study design should be highlighted.

6. The paper should report how privacy (data privacy) and confidentiality issues were handled during and after the zoom audio interview. Third-party Zoom handling of data and how the researchers and participants maintain privacy and confidentiality while using an audio interview should be reported.

Reviewer #2: Dear author,

It was a great privilege to have read your research article. The manuscript is well structured, with a strong introduction outlining the significance of marriage as a social institution and framing marital dissatisfaction as an underexplored but crucial area of study. The focus on the Nigerian immigrant experience in North America (NINA) adds a unique and relevant perspective in filling a significant gap in the literature. The study also incorporates rich qualitative data from participants, adding depth and perception to the discussion. Deriving from the discussion, the conclusion effectively summarizes the key findings of the study and provides actionable insights for policymakers, counselors, and married couples. Broadly, the study bridges the gap between research and practical applications.

Main Observations

Clarify marital dissatisfaction

I will suggest that you clearly define what constitutes marital dissatisfaction (e.g., routine stress versus abuse or emotional disconnection). For example, the information provided in line 536 suggest exhaustion of the participant rather than dissatisfaction in their marriage. Ensure consistency between the conceptual framework and participant responses to strengthen the study’s focus.

Expand participant details

The methodology adopted for the study is robust and thoughtfully considered. However, it would be beneficial to include participants’ occupations and their personal experiences with marital dissatisfaction. Clarify whether participants are direct experiencers or observers (e.g., counselors, clergy), as this impacts the credibility of their insights. This context would support the quotes used to illustrate the identified themes. The omission of these factors in the inclusion criteria seems to undermine the study’s findings, especially since you frequently reference the lived experiences of participants (lines 104, 115) and their observations (lines 239, 434).

Methodology

The study’s qualitative sample size is relatively small to determine the prevalence of marital dissatisfaction (line 136), particularly since none of the participants reported major issues in their marriage (Table 1). Without information on the selected participants’ marital dissatisfaction or their direct interactions with couples facing marital challenges, it is difficult to ascertain the contributory factors associated with marital dissatisfaction (line 137), especially if the participants neither experience marital dissatisfaction nor hold positions of authority such as marriage counsellors, marital legal advisers, or religious leaders who may have direct and open interaction with couples facing marital challenges.

However, these observed limitations may not be regarded as significant because the title of the article suggests that the study is on the 'perceived' causes of marital dissatisfaction. Nevertheless, for consistency, sentences that reference lived experiences and observations should be revised to align with the title of the article.

Discussion: Areas for improvement

While the study strongly critiques patriarchy structures and gender dynamics within Nigerian immigrant marriages, it does not sufficiently acknowledge cases where traditional values might contribute to marital stability, as you have noted in your earlier studies (Ade-Oshifogun, Jochebed B. et al, 2019). A more balanced discussion that incorporates instances of successful cultural integration or adaptive strategies would strengthen the discussion, especially with regards to why, despite the observed factors associated with marital dissatisfaction, Nigerian marriages are still stable with a high degree of marital satisfaction as already established in your previous study.

Secondly, your discussion presents Nigerian men as predominantly resistant to change and women as universally oppressed or seeking empowerment. While these trends may be common, acknowledging variations (such as men who adapt egalitarian norms or women who prefer traditional roles) would provide a more nuanced discussion. Furthermore, the claim in lines 414 to 419 that Nigerian women are more likely to secure better paying jobs than their husbands require substantiation from relevant literature. Is there any explanation for the differentials – is it that Nigerian women have superior qualifications and more experience than their male counterpart? Indeed, various reports have suggested that the median Nigerian household income is higher than the total population’s in the United States (United States Censors Bureau, 2023). Such information needs perceptive discussion within the overall framework of the study.

Repetitive points and redundancy

Some themes, particularly norms and gender role conflicts, are reiterated multiple times in similar ways. Consider these points for conciseness to improve readability and maintain reader engagement.

Conclusion

Overall, your manuscript presents a compelling and well-researched discussion on marital dissatisfaction among Nigerian migrants in North America. Its strengths lie in its cultural analysis, participant insights, and practical implications. However, a more balanced perspective, more information on participants, and a brief articulation of the concept of marital dissatisfaction would enhance the depth of the research. Addressing these areas would result in a more comprehensive and nuanced understanding of the topic.

Best wishes.

6. PLOS authors have the option to publish the peer review history of their article (what does this mean? ). If published, this will include your full peer review and any attached files.

**Do you want your identity to be public for this peer review?** For information about this choice, including consent withdrawal, please see our Privacy Policy .

Reviewer #1: **Yes: ** Chiedozie G. Ike

Reviewer #2: **Yes: ** Dr. Alex Asakitikpi

---

## [Author Response · Author response to Decision Letter 1]

23 May 2025

The response to reviewers' comments is in the uploaded file titled "Response to Reviewers."

---

## [Decision Letter · Decision Letter 1]

18 Jun 2025

2/17/2025

Perceived causes of marital dissatisfaction among Nigerian immigrants in North America: A qualitative study

PONE-D-25-08718R1

Dear Dr. Ade-Oshifogun,

We’re pleased to inform you that your manuscript has been judged scientifically suitable for publication and will be formally accepted for publication once it meets all outstanding technical requirements.

Kind regards,

Patrick Ifeanyi Okonta, MBBCh, MPH, FWACS, FMCOG, MD, DRH

Academic Editor

PLOS ONE

Additional Editor Comments (optional):

Reviewers' comments:

Reviewer's Responses to Questions

**Comments to the Author**

1. If the authors have adequately addressed your comments raised in a previous round of review and you feel that this manuscript is now acceptable for publication, you may indicate that here to bypass the “Comments to the Author” section, enter your conflict of interest statement in the “Confidential to Editor” section, and submit your "Accept" recommendation.

Reviewer #1: All comments have been addressed

Reviewer #2: All comments have been addressed

2. Is the manuscript technically sound, and do the data support the conclusions?

Reviewer #1: Yes

Reviewer #2: Yes

3. Has the statistical analysis been performed appropriately and rigorously? 

Reviewer #1: Yes

Reviewer #2: N/A

4. Have the authors made all data underlying the findings in their manuscript fully available?

Reviewer #1: Yes

Reviewer #2: No

5. Is the manuscript presented in an intelligible fashion and written in standard English?

Reviewer #1: Yes

Reviewer #2: Yes

6. Review Comments to the Author

Reviewer #1: The authors have addressed all the comments. However, to publish the manuscript, the manuscript should be reviewed for repetition of sentences any other grammar errors that might have escaped this round of review. For example. The first paragraph under the 'discussion on cultural factors' line 471-478 (see the tracked changed version below on the same section). The sentence is repeated. Under the 'Discussion on Job Situation' section, line 598-603, 621-625. (please see also the version below to delete the repeated sentence), we find the same error. See also the error in the section 'Discussion of lack of relationship skills.', 746-748.

Reviewer #2: Dear author,

It was my pleasure reading your manuscript again, and the insights it revealed regarding participants' perceptions of causes of marital dissatisfaction among immigrants in North America. I am quite pleased with the thoroughness of the study and the discussion that follows, demonstrating a good understanding of the multifaceted factors underlying marital dissatisfaction among the study population. It would be more revealing if a follow-up study could be conducted by selecting individuals who have undergone or are undergoing marital problems (such as divorce or separation for example) to obtain first-hand information for the causes of their situation. I hope you will have the time and resources to conduct further research in this field.

I look forward to reading more of your work.

Best wishes.

7. PLOS authors have the option to publish the peer review history of their article (what does this mean? ). If published, this will include your full peer review and any attached files.

**Do you want your identity to be public for this peer review?** For information about this choice, including consent withdrawal, please see our Privacy Policy .

Reviewer #1: **Yes: ** Chiedozie Godian Ike

Reviewer #2: **Yes: ** Dr. Alex Asakitikpi

---

## [Editor Report · Acceptance letter]

PONE-D-25-08718R1

PLOS ONE

Dear Dr. Ade-Oshifogun,

I'm pleased to inform you that your manuscript has been deemed suitable for publication in PLOS ONE. Congratulations! Your manuscript is now being handed over to our production team.

Kind regards,

on behalf of

Professor Patrick Ifeanyi Okonta

Academic Editor

PLOS ONE